# Then There Were Plenty-Ring Meristems Giving Rise to Many Stamen Whorls

**DOI:** 10.3390/plants10061140

**Published:** 2021-06-03

**Authors:** Doudou Kong, Annette Becker

**Affiliations:** Institute of Botany, Justus-Liebig-University, Heinrich-Buff-Ring 38, 35392 Gießen, Germany; Doudou.Kong@bot1.bio.uni-giessen.de

**Keywords:** floral meristem, polystemony, numerous stamens, evo–devo, ring meristem

## Abstract

Floral meristems are dynamic systems that generate floral organ primordia at their flanks and, in most species, terminate while giving rise to the gynoecium primordia. However, we find species with floral meristems that generate additional ring meristems repeatedly throughout angiosperm history. Ring meristems produce only stamen primordia, resulting in polystemous flowers (having stamen numbers more than double that of petals or sepals), and act independently of the floral meristem activity. Most of our knowledge on floral meristem regulation is derived from molecular genetic studies of *Arabidopsis thaliana*, a species with a fixed number of floral organs and, as such of only limited value for understanding ring meristem function, regulation, and ecological value. This review provides an overview of the main molecular players regulating floral meristem activity in *A. thaliana* and summarizes our knowledge of ring primordia morphology and occurrence in dicots. Our work provides a first step toward understanding the significance and molecular genetics of ring meristem regulation and evolution.

## 1. Introduction

Flowers are among the most beautiful examples of mutually beneficial relationships between animals and plants. Flowers are the major innovation of angiosperms, and their astonishing diversification is thought to be the major driver of the enormous success of the flowering plants. Even though most flowers are composed of only four different types of organs, their variability in structure, color, and scent is amazing and fine-tuned for attracting insect or bird pollinators or for allowing effective wind pollination [1,2,3].

In this review, we first outline the general morphogenesis of the floral meristem and describe the functions of the key genes involved in regulating maintenance and termination of the floral meristem. We then explain the concept of polystemony and identify lineages with independent origin of polystemony.

In many species, flower primordia arise from the flanks of the inflorescence meristem in a precise pattern (phyllotaxy), and, in few cases only, the shoot apical meristem turns into a floral meristem. For several species, the positioning of the flower primordia follows a localized auxin maximum via PIN1-mediated auxin transport in the epidermal L1 layer [4]. After initiation of the flower primordium, a meristem-to-organ boundary forms, separating the floral primordium from the rest of the plant body and the floral meristem enters its growth phase [4,5]. Auxin transport is now reversed, away from the primordium, to create an auxin sink in the central tissues of the shoot or inflorescences mediating the connection of the floral primordium to the vasculature [6].

In *Arabidopsis thaliana*, all floral organs arise from the floral meristem which, unlike the shoot or root meristem, undergoes a genetically defined succession of initiation, maintenance, and termination such that stem cell activity ceases once all floral organs are formed [5]. The floral meristem releases floral organ primordia at its flanks, and the number and position of stem cells is tightly regulated ensuring meristem stability, as well as allowing cells to accumulate before organogenesis. Thus, the meristem is a defined structure even though its constituent cells are constantly changing. Cells exiting the meristem adopt their future cell fate according to their position within the meristem, while remaining in their original cell layers. Floral organs arise as primordia, composed of undifferentiated cells, which adopt a floral organ identity, and they subsequently grow and differentiate into the floral organs. In most flowers, floral organs are arranged in whorls, with sepals being the outermost organs, the second whorl involving the petals, followed by the pollen grain-producing stamens, and, in the center of the flower, the gynoecium is formed, harboring the ovules. After fertilization, the ovules develop into seeds, and the gynoecium is transformed into the fruit [7]. The development of floral organs follows a genetically determined blueprint, which was elucidated by molecular genetics studies in model species such as *Arabidopsis thaliana*, *Antirrhinum majus*, and grasses. Several recent reviews summarized our knowledge on floral organ identity, its variation and origin, and remaining open questions [8,9,10,11], which are not covered here.

Floral meristems are, thus, dynamic systems, balancing proliferation and termination in a species-specific manner, leading to a species-specific size and a number of different floral organ types. The shoot and inflorescence meristems are often and, to at least some extent, indeterminate, but stem cell maintenance in the floral meristem ends after the initiation of the gynoecium, rendering the floral meristem determinate [12]. *Arabidopsis thaliana* has a fixed number of floral organs and is typical for the canalized floral ground plan of many core eudicots. However, deviations from this ground plan are numerous, and little is known about the conservation of the gene regulatory network (GRN) balancing floral meristem action. While it seems reasonable to assume that some degree of conservation exists in floral meristem GRNs, this genetic system also allows extreme examples of variation of floral organ number (merosity), the floral meristem of the Ranunculaceae *Laccopetalum giganteum* generates up to 10,000 small carpels [13], while the bisexual Hydatellaceae (sister to all other Nymphaeales) flowers consist of either a single stamen or a single carpel [14].

## 2. Regulation of Stem Cell Activity in the Floral Meristem

The genetic networks regulating flower development have been characterized to the largest extent in the genetic model organism *Arabidopsis thaliana*. Molecular data from other species are largely missing; thus, we show the *A. thaliana* way of modulating floral meristem fate, keeping in mind that this floral meristem generates a fixed number of whorls and organs per whorl.

The regulation of floral meristem termination involves a highly complex regulatory interplay of transcription factors, microRNAs, receptor-like kinases and their ligands, MAP kinases, and chromatin remodeling (comprehensively reviewed in [15]); the homeodomain-containing transcription factors WUSCHEL (WUS) and SHOOT MERISTEMLESS (STM) function synergistically during shoot, inflorescence, and floral meristem development and maintenance. In *wus* and *stm* mutants, the floral meristem ceases prematurely because a smaller number of stem cells in the central zone is formed, which cannot keep up with the production of a larger number of cells in the periphery required for floral organ formation [16,17,18,19]. Conversely, overexpression of *WUS* leads to a conversion of cells in organ primordia into cells with stem cell characteristics [20]. *WUS* activation is limited by the CLAVATA (CLV) signaling loop, where the small apoplastic signaling peptide CLV3, activated by WUS in L1 and L2, moves toward L3 where it binds to the CLV1 and CLV2 receptors and other receptor-like kinases to indirectly repress *WUS* expression [21]. When any of the *CLV* genes is nonfunctional, the meristem’s central zone increases due to an overexpression of *WUS* [22,23]. WUS also activates the floral homeotic gene *AGAMOUS* (*AG*) by direct binding to its second intron; thus, *WUS* not only regulates stem cell maintenance but is also required for floral organ initiation [24]. Once the floral meristem has produced all floral organ primordia, it is consumed in the process of carpel development, and stem cell activity ceases. For this process, the floral homeotic C function protein AG is required to repress *WUS* by directly binding to its genomic locus and by the recruitment of polycomb group proteins that methylate chromatin to silence the *WUS* genomic locus [19,25]. In addition, AG activates *KNUCKLES (KNU)* by replacing repressing histone marks with activating ones in a time-dependent manner, and *KNU* then represses *WUS* expression [26]. Moreover, the floral homeotic A function gene *APETALA2* (*AP2*), which also promotes stem cell fate in the floral meristem by directly antagonizing *AG*, is translationally repressed by miR172d to achieve floral determinacy [27,28]. Furthermore, *AP2* not only negatively regulates the expression of *AG*, but also independently regulates the activity of floral meristem [29]. At the same time, there is evidence that *AP2* negatively regulates the *CLV* signaling pathway [30].

Moreover, additional transcription factors, such as SUPERMAN (SUP), CRABS CLAW (CRC), PERIANTHA (PAN), and ULTRAPETALA (ULT), also regulate floral meristem maintenance and termination, but to a lesser extent and in a partially redundant way, suggesting a high level of genetic robustness in the floral meristem termination network [31,32,33].

More recently, auxin signaling was reported to play a major role in floral meristem termination on several levels. For example, SUP recruits chromatin remodelers to suppress auxin biosynthesis genes at early stages of floral meristem development. If this function is lacking, the floral meristem is enlarged and active for a prolonged time [34]. The AG target *CRC* connects floral meristem termination with gynoecium development by activating the putative auxin transporter TORNADO2 to promote gynoecium formation while the floral meristem terminates [35]. The auxin response factor ARF3 is regulated by auxin levels, as well as by *AG* and *AP2*, and it is required to terminate *WUS* activity in the floral meristem [36]. However, it remains unclear how auxin biosynthesis and transport contribute to floral meristem termination on the molecular level.

While several meristem regulators in other species such as *Solanum lycopersicum* (tomato), *Oryza sativa* (rice), and *Zea mays* (corn) have already been identified and functionally characterized, they all turned out to be orthologs of the genes analyzed in *A. thaliana* [37] except for *BEARDED-EAR* and *MOSAIC FLORAL ORGANS*1 from corn and tomato [38,39], respectively. Interestingly, many components of the floral organ identity and floral meristem termination regulatory network are conserved among seed and even non-seed plants. For example, *WUS* homologs are required for the initiation of cell growth during stem cell formation in the moss *Physcomitrella patens* [40], and several *WUS* homologs were also identified in gymnosperms [41].

## 3. How to Generate Multiple Stamen Primordia

Considering variations in the floral ground plan, the number of stamens is especially labile, and we concentrate on this trait in this section. The flower of the most recent common ancestor (MRCA) of all angiosperms most likely had more than 10 stamens arranged in three stamens per whorl, suggesting at least four whorls generating stamen primordia [42]. In recent species, the most common stamen arrangement in monocots and core eudicots involves two whorls with the outer stamens in alternate position to the petals [43]. In the ANITA grade, whose members are sister to all other angiosperms, stamen numbers are especially labile and range from one in Hydatellaceae and Chloranthaceae to up to 300 in Schisandraceae [43]. In species with an increased number of stamens, these are often not arranged spirally, but rather an androecial ring primordium forms, generating whorls of stamens. The direction of stamen primordia initiation can be toward the center of the flower (centripetal), toward the periphery (centrifugal), or bidirectional (Figure 1). The ring primordium sometimes subdivides into sectors and is, thus, fragmented [43]. This additional ring meristem provides a mean to uncouple stamen initiation from carpel initiation, such that the last stamens initiate after the carpels are already far advanced in their development (Figure 1).

## 4. Ring Meristems in Eudicots

Eudicots comprise 44 orders, and ring meristems producing extra stamen whorls are found in 13 of them (APG, 2016 and Figure 2) [45]. Interestingly, the fraction of families with ring meristems per order is unequally distributed across the phylogeny. In Ranunculales, all families have members with ring meristems, and almost all combinations of closed/fragmented architecture and directions of stamen primordia initiation occur. Asterales consist of 11 families and a staggering number of almost 27,000 species, but include only the genus *Taraxacum* within the Asteraceae that has a ring meristem. In most orders, only a fraction of members develop ring meristems. For example, in Proteales, Fabales, Brassicales, and Myrtales, only a single family includes members with ring meristems. Interestingly, most ring meristems are of the closed type, and the number of orders with centripetal and centrifugal initiation of primordia is similar. However, bidirectional initiation is rarely found (Figure 2). Interestingly, centripetal primordia initiation is predominant in the non-core eudicot orders of Ranunculales and Proteales; however, in the core eudicots, centrifugal initiation is found more often. In several groups, a closed ring meristem is associated with centripetal primordia initiation, for example, in the Ranunculales, Protetales, Fabales, Hamamelidaceae, and Portulacaceae. In contrast, all combinations of primordia initiation direction and closed/fragmented ring meristem occur within the Malvales. According to our data, there seems to be no preference in the co-occurrence between closed and fragmented ring meristems and the direction of stamen primordia initiation. However, data compilation on the species level are lacking so far.

One of the well-known floral abnormalities is represented by double-flower phenotypes. These develop extra petals, most often at the expense of stamens, and are selected for in many ornamental species of high economic value. Interestingly, this phenomenon often correlates with mis regulation of *AG* orthologs or *APETALA2*-like genes [46], and a modification of ring meristem activity was not reported in these cases.

## 5. Are Ring Meristems in Dicots of Independent Origin?

In the Proteales and Ranunculales, which are sister to all other eudicots, ring meristems are found in all (Ranunculales) or at least some (Proteales) families; however, in Buxales and Trochodendrales, also non-core eudicots, ring meristems are missing. It, thus, remains unclear if the MRCA of eudicots had a floral ring meristem, and, if this was the case, we need to propose losses of ring meristems in the majority of eudicot orders. Generating more stamen and, in consequence, more costly pollen grains require a selective advantage for trait maintenance. If a high stamen number is not advantageous, independent losses may be rather frequent. In contrast, it seems also reasonable to propose that the eudicot MRCA had no ring meristem; however, according to this hypothesis, at least 13 independent gains of ring meristem activity need to be postulated. Nevertheless, the diversity of ring meristem architectures in dicots may be a hint toward several independent gains, as an evolutionary trend from an ancestral to a more derived condition in ring meristem structure cannot even be safely derived in the Ranunculales.

## 6. Conclusions and Outlook

Here, we compiled recent findings on transcription factors, signaling pathways, and phytohormone action, all involved in floral meristem termination of *Arabidopsis*. We further demonstrated that at least some components of the floral meristem regulatory systems are conserved between eudicots and mosses. However, the molecular regulation of floral meristem termination has mainly been analyzed in *Arabidopsis*, and even knowledge on this process in the monocot model species rice is lagging behind. Meristematic activity regulation in species with ring meristems has, to our knowledge, not been studied so far, leaving many questions open. This raises the question related to how the meristem activity in the central floral meristem and the ring meristem is differentiated and correlated in space and time, when presumably at least partially similar developmental mechanisms govern this activity. Unlike the ring meristem, which gives rise to only stamen primordia, the floral meristem releases lateral organs whose order and identity are precisely regulated. In most species, termination of the floral meristem activity is coupled to gynoecium initiation. However, ring meristems give rise to several whorls of stamen, and it remains unclear how these meristems are initiated, separated from the floral meristem, and then terminated. Furthermore, in addition to open questions related to their developmental regulation remaining unanswered so far, their ecological significance is unclear. For example, it would be interesting to test whether ring meristem occurrence and polystemony are correlated to specific pollination modes and pollinator preferences or how fast polystemony disappears when selfing becomes more prominent. Moreover, ring meristems producing varying numbers of stamen may shift the balance between male and female contributions toward sexual reproduction. Thus, the regulation of ring meristem activity termination, at least in some species [44], may be more flexible than that of the central floral meristem. For example, it may depend on environmental cues, on the accumulation of sufficient reserves, or on other metabolic changes during a plant’s maturation process.

Learning about the mechanisms generating and regulating ring meristems requires new genetic model species with ring meristems, if possible, closely related to those without a ring meristem, for example, from the Ranunculales family. This allows comparative transcriptomics of flowers that develop ring meristems with those that do not. In addition, ring meristem tissue may be analyzed separately from the central floral meristem to reveal genes differentially expressed between ring and central floral meristems. The role of these candidate genes in ring meristem establishment and regulation may then be revealed by knockout or knockdown approaches.

Thus, our review provides only a first glance into a morphological structure that is enigmatic in terms of the molecular mechanism of its regulation and with respect to its origin, evolution, and ecological significance.

## Figures and Tables

**Figure 1 plants-10-01140-f001:**
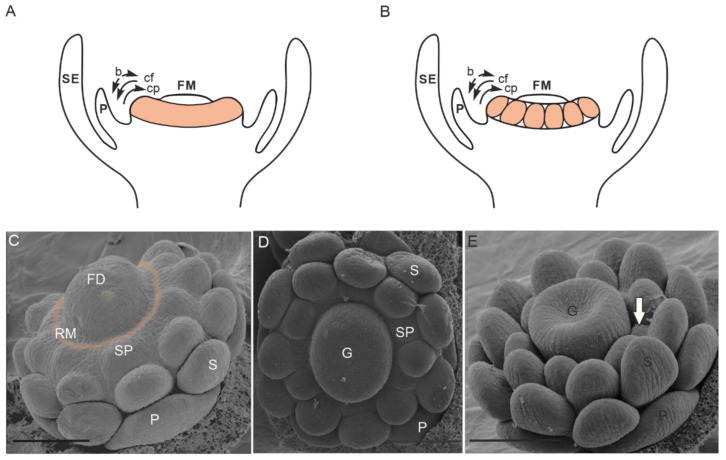
Flowers with ring meristems: (**A**) closed ring meristem and (**B**) fragmented ring meristem; the ring meristem is shown in orange. Black arrows indicate the direction of stamen primordia initiation. (**C**–**E**) Ring meristem of *Eschscholzia californica* forming stamen primordia. (**C**) Bud of a late stage 4 flower in which the floral dome separates from the ring meristem (indicated by a blurred orange line). (**D**) Stage 5 bud showing the continuous formation of stamen primordia while the gynoecium has already initiated. (**E**) Stage 6 bud showing the formation of new stamen primordia (white arrow) at the time when the early stamens already form a flat surface for microsporangia initiation. The sepals were removed in all images, and staging was done according to Becker et al. (2005) [44]. Abbreviations: b, bidirectional; cp, centripetal; cf, centrifugal; FD, floral dome; FM, floral meristem; G, gynoecium; P, petal; RM, ring meristem; SE, sepal; SP, stamen primordium; S, stamen. Scale bar in C = 86 µm, in D = 100 µm, in E = 120 µm ((**A**,**B**) modified from Endress, 2011; (**C**–**E**) from Becker, 2016 with permission) [7,43].

**Figure 2 plants-10-01140-f002:**
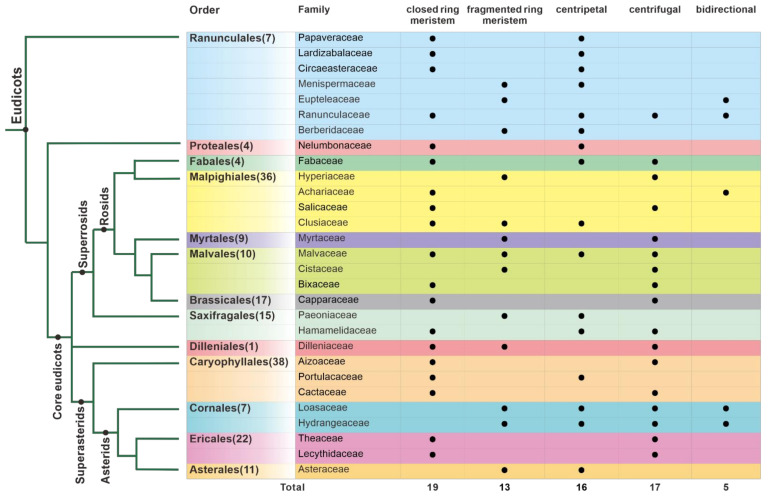
Distribution and types of ring meristems across eudicots. On the left, a simplified phylogeny shows only orders and families with ring meristem occurrence. Numbers in brackets next to the orders indicate the number of families in the respective order. On the right, the type of ring meristem is indicated. Numbers below summarize occurrences of morphological traits. Based on [43,45,47,48,49,50,51,52,53,54,55,56,57].

## Data Availability

Not Applicable.

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
