# Peer review of "Then There Were Plenty-Ring Meristems Giving Rise to Many Stamen Whorls"

_plants, 2021, doi:10.3390/plants10061140_

Round 1

Reviewer 1 Report

This is a well written and concise review of ring meristems and their evolution. It is further enhanced by helpful and attractive figures. I have no qualms about recommending this article for publication.

I have only one query - whether a period or a comma should be used for designating thousands. Here a period is used - which more frequently infers decimal places than "thousands". There are two instances:

p 2 line 27

p4 line 6

Author Response

Thank you for your very positive review of our manuscript. We have
exchanged the period with the comma on pages 2 and 4.

Reviewer 2 Report

The review sufficiently covers the modern progress in the molecular control of meristem maintenance and rise the problem of ring meristem activity in different clades of Angiosperms. It is a worthy to be published.

The only misprint I could find is in figure 1 legend. 

(C-E) Ring meristem of Eschscholzia californica - E also should be attributed to E. californica.

FM, floral meristem; ... P, petal - capitals.

Scale bar in A (C?) = 86 μm, in B (D?) = 100 μm, in C (E?) = 120 μm

A & B are schematic presentations without scale bars. Check it, please.

As a matter of discussion, it would be interesting, whether ring meristem is responsible for whorl multiplication in double varieties of cultivated ornamentals. These observations may broaden the list of families with [potentially active] ring meristem.

Author Response

Thanks for your very positive review of our manuscript. We have followed all your suggestions regarding the figure 1 legend and corrected all errors. Regarding the whorl multiplication in double varieties of cultivated ornamentals: to our knowledge there is literature about double flower resulting from floral homeotic mutations in e.g. C class genes converting stamens into petals, but this is not directly related to prolonged meristem activity. We have added an explanatory sentence and a reference on roses to chapter 4.

Reviewer 3 Report

This review discussed the phenomenon of ring meristems, which give rise to many stamen whorls. It is indeed an interesting observation and worths more attention. The 

I have two main concerns.

  1. Part 2 of the review is text heavy. It will help if there is a figure to illustrate the regulation control. Also, the connection between this part and the later ring meristem part is not very obvious. I would like to see some discussion about the potential mechanisms that generate ring meristem.
  2. The review pose an interesting question, but did not offer a solution. It would be nice if the authors can point out some potential future directions, e.g. which organism might be suitable to answer this question, what approach should be used, etc.

Author Response

We agree that part 2 deserves a figure. However, there are a number of well written reviews on the regulation of floral meristem activity in Arabidopsis
including high quality figures. We tried to come up with our own figure on this but found ourselves merely copying from previous publications. We thus prefer to refer to a comprehensive review on the topic of floral meristem activity regulation and added the reference to the text. Unfortunately, nothing is published so far on the mechanisms that generate the ring meristem.
While this topic is in the center of one of my recent research projects, we find nothing going beyond the morphological description of ring meristems in the literature. Thus, our thoughts on this would be merely speculative.
Regarding the second concern: we have added our experimental approach to the last section of the manuscript as a suggestion on how we could learn more about the molecular mechanisms involved in ring meristem formation and regulation.